# Phlogopite-Reinforced Natural Rubber (NR)/Ethylene-Propylene-Diene Monomer Rubber (EPDM) Composites with Aminosilane Compatibilizer

**DOI:** 10.3390/polym13142318

**Published:** 2021-07-14

**Authors:** Sung-Hun Lee, Su-Yeol Park, Kyung-Ho Chung, Keon-Soo Jang

**Affiliations:** Department of Polymer Engineering, School of Chemical and Materials Engineering, The University of Suwon, Hwaseong 18323, Korea; atlasba@naver.com (S.-H.L.); tnduf3971@naver.com (S.-Y.P.)

**Keywords:** phlogopite, natural rubber (NR), ethylene-propylene-diene monomer rubber (EPDM), phlogopite, mechanical properties, compatibility

## Abstract

Rubber compounding with two or more components has been extensively employed to improve various properties. In particular, natural rubber (NR)/ethylene-propylene-diene monomer rubber (EPDM) blends have found use in tire and automotive parts. Diverse fillers have been applied to NR/EPDM blends to enhance their mechanical properties. In this study, a new class of mineral filler, phlogopite, was incorporated into an NR/EPDM blend to examine the mechanical, curing, elastic, and morphological properties of the resulting material. The combination of aminoethylaminopropyltrimethoxysilane (AEAPS) and stearic acid (SA) compatibilized the NR/EPDM/phlogopite composite, further improving various properties. The enhanced properties were compared with those of NR/EPDM/fillers composed of silica or carbon black (CB). Compared with the NR/EPDM/silica composite, the incompatibilized NR/EPDM/phlogopite composite without AEAPS exhibited poorer properties, but NR/EPDM/phlogopite compatibilized by AEAPS and SA showed improved properties. Most properties of the compatibilized NR/EPDM/phlogopite composite were similar to those of the NR/EPDM/CB composite, except for the lower abrasion resistance. The NR/EPDM/phlogopite/AEAPS rubber composite may potentially be used in various applications by replacing expensive fillers, such as CB.

## 1. Introduction

Rubbers with high diene contents, such as natural rubber (NR), polybutadiene, nitrile rubber, and styrene-butadiene rubber, exhibit poor outdoor properties [1,2,3,4]. In particular, NR is a natural biosynthesis polymeric rubber composed of isoprene with minute organic impurities and water. Among the high diene concentration-containing rubbers, NR has been extensively exploited, either alone or in combination with other materials, in a wide range of applications, such as in automobiles, trains, tires, conveyor belts, hoses, balls, cushions, gloves, and shoes [1,5,6,7]. NR features good elasticity (resilience), strength, and processing characteristics, and excellent physical and mechanical properties [8,9,10,11]. However, its poor heat, UV, oxygen, and ozone resistance caused by the highly unsaturated polymeric backbone hinders some applications requiring weathering resistance [12,13,14].

Instead of creating a new singular rubber material, an alternative facile avenue of developing a new advanced material with the required properties is the blending of different rubbers [15,16]. Ethylene-propylene-diene monomer (EPDM) rubber is a less unsaturated elastomer and is polymerized using ethylene and propylene, with a small concentration of nonconjugated diene. EPDM features good aging characteristics, good resistance to weathering, and oxidation and chemical resistance [17,18]. The poor environmental (ozone, heat, and UV) resistance of the NR phase was considerably countered by the EPDM phase without sacrificing the unique properties of NR in NR/EPDM rubber blends [19,20,21]. EPDMs exhibit a balance among chemical, electrical, thermal, and mechanical properties [22]. Thus, NR/EPDM rubber blends have been extensively employed in various applications, particularly in tire-related parts [23]. However, the difference in the olefin content between NR and EPDM causes their respective cure rates to be incompatible, which adversely affects the mechanical properties of the blends [24,25,26]. The mechanical properties have been enhanced by many approaches such as epoxidizing NR, grafting a vulcanization inhibitor, grafting an accelerator onto EPDM (modified EPDM), two-stage vulcanization, reactive blending, and incorporating compatibilizers or reinforcing fillers [27,28,29,30,31].

In terms of reinforcing fillers, carbon black (CB), silica, clay, and CaCO_3_ are widely utilized in rubber systems, similar to thermoplastic polymer systems [32,33,34,35]. In particular, NR/EPDM blend systems inevitably require fillers to enhance the mechanical properties, improve processability, add colors, and reduce the cost. The incorporation of fillers into rubbers brings about diverse interactions at the rubber–filler interfaces [36,37]. Among the various fillers, CB and silica are the most widely used [20,38]. Despite their strong interactions with rubbers, they are relatively expensive, and the application of CB has been hindered owing to its black color [37,38]. Therefore, potential fillers such as soybean protein [39], organo-montmorillonite [40], bio-based fibers [41,42], and biochar [43] have been investigated as alternatives.

Mica (dioctahedral: muscovite and paragonite; trioctahedral: biotite and phlogopite) is commonly utilized to improve the mechanical properties, dynamic characteristics, wear resistance, and processability of rubber composites [44,45,46,47]. Among mica fillers, phlogopite as a filler has been introduced in applications such as adhesives [48], plastic parts [49,50,51], and cosmetics [52]. However, phlogopite has not been employed in rubber applications. Thus, in this study, the effects of phlogopite and a coupling agent for the filler–rubber interfaces on mechanical properties were explored and compared with those of carbon black and silica.

## 2. Experiment

### 2.1. Materials

Natural rubber (NR, STR 5L) and ethylene-propylene-diene monomer rubber (EPDM, Keltan KEP-960N(F)) were purchased from PAN STAR Co. (Bangkok, Thailand) and Kumho Polychem Co. (Seoul, Korea), respectively. Carbon black (CB; N330, 28–36 nm, Aditya Birla Chemicals Co., Estado indio, India), phlogopite (LKAB Minerals Co. 40–80 μm, Luleå, Sweden), and silica (3M Co. 20–30 μm, St. Paul, MN, USA) were used as reinforcing fillers. Zinc oxide (ZnO), stearic acid (SA), tetramethylthiuram disulfide (TMTD), N-cyclohexyl-2-benzothiazyl sulfonamide (CBS), and sulfur were purchased from Puyang Willing Chemical Co. (Puyang, China). Aminoethylaminopropyltrimethoxysilane (AEAPS, OFS-6020, Dow Chemical Co., Midland, MI, USA), DI water (BNOChem Co., Cheongju, Korea), and acetic acid (BNOChem Co., Cheongju, Korea) were used for improving the compatibility of the NR/EPDM blend and phlogopite. The AEAPS solution (AEAPSS) was composed of AEAPS, DIW, and acetic acid (30:20:50 wt%).

### 2.2. Rubber Compounding

The pre-mixing (mastication) of NR (62.5 wt%) and EPDM (37.5 wt%) was carried out using an open two-roll mill to produce a blend band form. Fillers (10 parts per hundred resin (phr)) were added to the rubber blend, followed by the incorporation of ZnO (5 phr) and stearic acid (2 phr). In the case of phlogopite addition, AEAPSS (0, 2, 5, and 10 phr) was added together with phlogopite. After the mixture was mixed for 15 min, TMTD (1 phr), CBS (1 phr), and sulfur (1 phr) were added to the mixture and mixed for 10 min.

### 2.3. Curing Characteristic

#### 2.3.1. Cure Time (T_90_)

Cure characteristics were examined at 170 °C using a rheometer (DRM-100, Daekyung Engineering Co., Ulsan, Korea) to determine the T_90_ (time at 90% cure extent). The mean values were determined on the basis of five measurements for each sample.

#### 2.3.2. Mooney Viscosity

The Mooney viscosity was determined at 125 °C for 4 min using a Mooney viscometer (DWV-200C, Daekyung Engineering Co., Ulsan, Korea). The sample was pre-heated at 125 °C for 1 min prior to the measurements.

### 2.4. Mechanical Properties

#### 2.4.1. Tensile Properties

Uniaxial tensile deformation was performed using a universal testing machine (UTM; DUT-500CM, Daekyung Engineering Co., Ulsan, Korea). The tests were performed according to ISO 37. The cross-section of the specimen was 6 × 2 mm, and the gauge length was 40 mm. The specimens were elongated at a constant strain rate of 500 mm/min at 22–24 °C. The mean values were determined based on five specimens. The toughness was determined by integration of stress–strain curves.

#### 2.4.2. Hardness

The shore A hardness of the rubber blends and composites was measured according to ISO 48 using a hardness tester (306L, Pacific Transducer instruments, Los Angeles, CA, USA). The mean values were determined based on five specimens.

#### 2.4.3. Rebound Resilience

The rebound elasticities of the rubber blends and composites were measured using a ball rebound tester according to ISO 4662. The specimens were maintained at 22–24 °C for 2 h prior to the measurements. The round ball fell onto the samples, and the rebounding height was measured. The mean values were determined based on five specimens.

#### 2.4.4. Abrasion Resistance

An abrasion resistance test was conducted using an abrasion tester (DRA-150, Daekyung Engineering Co., Ulsan, Korea). A 2.5 × 2.5 cm sample was placed on a cylindrical tester with a diameter of 15 cm. A cylindrical hammer (470 g) was applied to the sample surface to provide uniform contact forces in abrasive paper of 40 grit (XW341, Deerfos Co., Incheon, Korea). The sample was abraded by rotating it 200 times at 40 rpm. The pristine NR/EPDM was used as a reference sample. The abrasion resistance index (ARI) was calculated based on Equation (1).
(1)$$\mathrm{ARI}=\frac{\Delta m_{r}\;p_{t}}{\Delta m_{t}\;p_{r}}\times 100$$
where Δ*m_r_*, *p_r_*, Δ*m_t_*, and *p_t_* are the mass loss of the reference compound, density of the reference compound, mass loss of the test rubber, and density of the test rubber, respectively.

#### 2.4.5. Morphology

The morphologies of the NR/EPDM blend and NR/EPDM/phlogopite composites were observed by scanning electron microscopy (SEM; Apreo, FEI Co., Hillsboro, OR, USA) at an electron beam voltage of 10.0 kV (at the Center of Advanced Materials Analysis, University of Suwon, Hwaseong, Korea). The surface fractured during tensile tests was coated with a 5–10 nm-thick gold layer using a sputter coater prior to the SEM measurements.

#### 2.4.6. FTIR-ATR

Fourier transform infrared (FTIR) spectroscopy (Spectrum Two, PerkinElmer Inc., Waltham, MA, USA) with attenuated total reflection (ATR) mode was performed to investigate the AEAPSS treatment. The thickness of the cured sample was 1 mm. The scan number was 16.

## 3. Results and Discussion

Mooney viscosity tests have been widely utilized to measure the viscosity of raw rubber materials prior to vulcanization. The Mooney viscosity of the pristine NR/EPDM blend was the highest, whereas the incorporation of fillers (silica, CB, and phlogopite) into the blends reduced the Mooney viscosity (Figure 1). The low concentration of filler barely influences the Mooney viscosity. In particular, mica-based fillers with a platy architecture commonly decrease the Mooney viscosity [45,46]. The infiltration of AEAPSS 2 phr into the NR/EPDM/phlogopite elastomeric composites led to compatibilizing effects, thereby increasing the Mooney viscosity. However, after the threshold quantity of 2 phr, the Mooney viscosity decreased with the increasing AEAPSS concentration owing to the plasticization of the excess AEAPSS.

Rubber blends and composites with different formulations exhibited different curing behaviors. The optimum curing time for rubber materials is typically defined as T_90_, which is the time required for the torque to reach 90% of the maximum torque during curing. T_90_ is related to the time required for the development of the optimum properties. The curing behaviors and T_90_ were noticeably influenced by the filler incorporation because of the filler–rubber matrix interactions. The maximum torque for filler-reinforced rubbers typically decreases as a function of temperature. Figure 2 shows the T_90_ values of the NR/EPDM blends and composites with different fillers. The T_90_ values of the NR/EPDM/silica and NR/EPDM/CB composites were lower than those of the pristine NR/EPDM blend, whereas the incorporation of phlogopite into the blend without AEAPSS increased the T_90_ value. The NR/EPDM/phlogopite/AEAPSS composites showed the lowest T_90_ values among the other filler-embedded NR/EPDM composites. This indicates that the amine moieties of AEAPS accelerated the reaction rate of vulcanization [53,54].

Among various properties, the mechanical properties of rubber composites are crucial, especially for tire and automobile applications. For instance, the tensile strength exhibits the maximum stress before material failure in those applications. Figure 3a shows the tensile strengths of the NR/EPDM blend and composites. The incorporation of silica and CB into the NR/EPDM blend improved the tensile strength, whereas the tensile strength of the NR/EPDM/phlogopite composite barely changed, compared with that of the pristine NR/EPDM blend. This indicates incompatibility between the phlogopite filler and rubbers. The additional infiltration of AEAPSS into the NR/EPDM blend gradually enhanced the tensile strength of the composites as a function of the phlogopite concentration owing to the compatibilizing effect. The elongation at break of each composite except 2 phr AEAPSS was higher than that of the neat NR/EPDM blend, as shown in Figure 3b. The concentration of 2 phr AEAPSS was insufficient to coat the phlogopite. In the absence of SA (red box in Figure 3a–d), the compatibilizing effect was reduced, thereby resulting in a decrease in mechanical properties. The tensile moduli of NR/EPDM/silica and NR/EPDM/CB showed little change, whereas NR/EPDM/phlogopite, even without AEAPSS, portrayed a slight enhancement in the tensile modulus, as shown in Figure 3c. Analogous to the tensile strength results, the toughness of the composites was higher than that of the neat blend. The compatibilized NR/EPDM/phlogopite/AEAPSS 10 phr composite exhibited a higher toughness than the NR/EPDM/CB composite (Figure 3d). The stress–strain curves are shown in Appendix A. 

The hardness of rubbers typically indicates resistance to localized plastic deformation that is triggered by mechanical indentation (or abrasion). The hardness is determined by a combination of several factors, such as the elastic stiffness, strength, elongation, toughness, ductility, and viscoelasticity. The shore A hardness is routinely utilized for testing rubber materials. Each of the composites containing each filler exhibited greater hardness than the NR/EPDM blend, as shown in Figure 4. The incorporation of AEAPSS into the NR/EPDM/phlogopite composite slightly increased its hardness. The abrasion resistance indexes (ARIs) of the blends and composites are shown in Figure 5. The ARI of the NR/EPDM/CB composite was lower than that of the neat blend, whereas the incorporation of silica and phlogopite into the blend enhanced the ARIs. Among the various properties, the NR/EPDM/phlogopite composites showed the highest values of the ARI, compared with other composites containing silica or CB.

The elasticity and flexibility of the rubber polymer chains can be confirmed by the rebound resilience tests. The rebound resilience is defined as the ratio of the energy released by the deformation recovery to that required to generate the deformation. It is common for the rebound resilience of rubbers to decrease with increasing filler concentration. The infiltration of fillers into the rubbers reduces the elasticity of the rubber chains, thereby decreasing the resilience properties. Figure 6 shows that the NR/EPDM/silica composite exhibited the lowest reduction in rebound resilience. The effects of CB and phlogopite on the rebound resilience were analogous to each other.

Morphological studies of rubber composites are routinely performed by SEM. The morphologies of pure fillers (CB, silica, and phlogopite) and the dispersity of fillers in NR/EPDM systems are shown in Appendix A, respectively. Figure 7 shows the fractured surfaces of the NR/EPDM blend and composites. The NR/EPDM blend showed a smooth surface with little phase separation, as shown in Figure 7a. The silica- and CB-embedded rubber composites showed good dispersion, with slight agglomeration (Figure 7b,c). The incorporation of AEAPSS into the NR/EPDM/phlogopite composites compatibilized the filler surfaces and rubbers, as observed in Figure 7d,e.

FTIR-ATR was utilized to investigate the effect of AEAPSS on the phlogopite surface treatments, as shown in Figure 8. The broad peaks between 450 and 520 cm^−1^ indicate Si–O and Mg–O for phlogopite. The peaks at 950–990 cm^−1^ and 690 cm^−1^ are ascribed to Si–O for phlogopite [55]. The peak at 1560 cm^−1^ contributing to amine for AEAPS that appeared as 10 phr AEAPSS was added [56,57]. In addition, a peak at 610 cm^−1^ that is associated with the bending vibration of Si–O–Si was observed, probably due to the formation of Si–O–Si between the phlogopite and silane of AEAPSS [58]. On the basis of these results, SA acted as a coupling agent between the rubbers and AEAPSS, whereas AEAPSS acted as a compatibilizing agent between phlogopite and SA, as shown in Figure 9. This interplay created a useful NR/EPDM/phlogopite composite with enhanced properties.

## 4. Conclusions

The effects of phlogopite on various properties (curing behavior, and mechanical, elastic, and morphological properties) of an NR/EPDM blend were examined by comparing silica- and CB-reinforced NR/EPDM composites. In addition, the compatibilizing effect of AEAPSS was investigated for the NR/EPDM/phlogopite composites to further improve the phlogopite-embedded NR/EPDM composite. The combination of SA and AEAPSS provided compatibilizing effects between rubbers and phlogopite. The incompatibilized NR/EPDM/phlogopite composite without AEAPSS showed poorer properties than the NR/EPDM/silica composite, whereas NR/EPDM/phlogopite compatibilized by AEAPSS along with SA was superior to NR/EPDM/silica in terms of most properties. Compared with the NR/EPDM/CB composite, the compatibilized NR/EPDM/phlogopite composite exhibited slightly enhanced or similar properties, except for abrasion resistance. Thus, NR/EPDM/phlogopite/AEAPSS composites may potentially be used in various applications instead of silica- and CB-reinforced NR/EPDM composites.

## Figures and Tables

**Figure 1 polymers-13-02318-f001:**
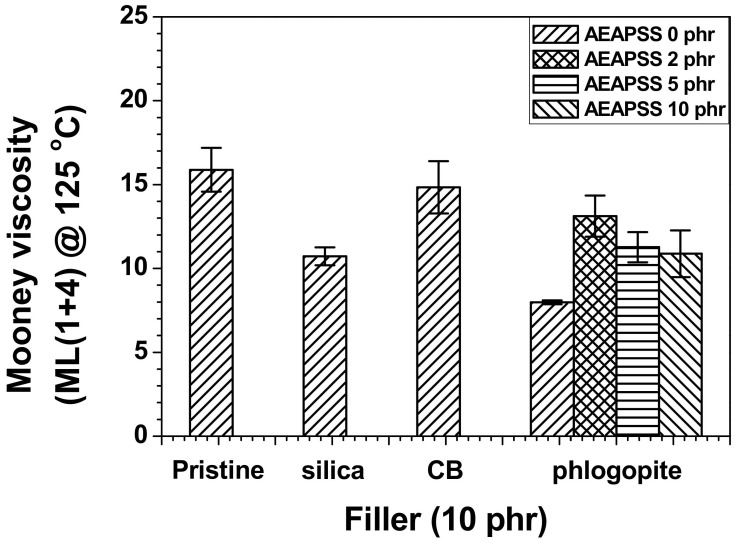
Mooney viscosities of the NR/EPDM blends and composites with different fillers and AEAPSS concentrations.

**Figure 2 polymers-13-02318-f002:**
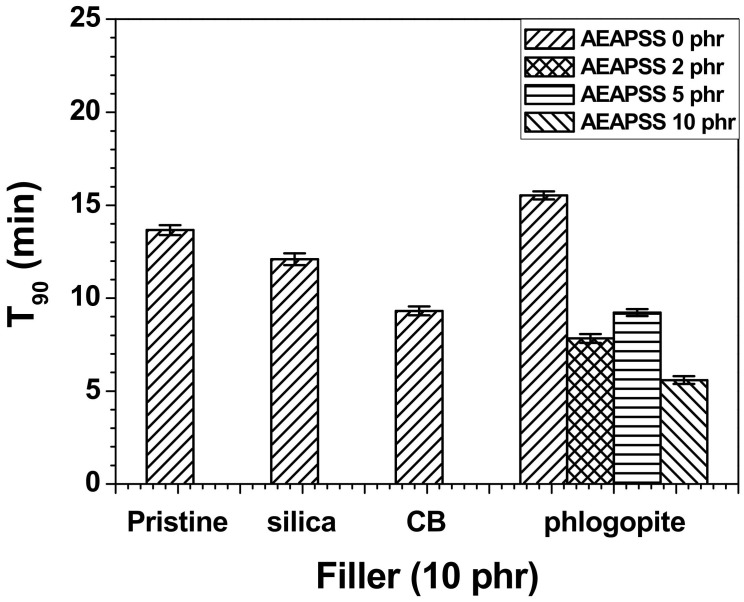
T_90_ of the NR/EPDM blends and composites with different fillers and AEAPSS concentrations.

**Figure 3 polymers-13-02318-f003:**
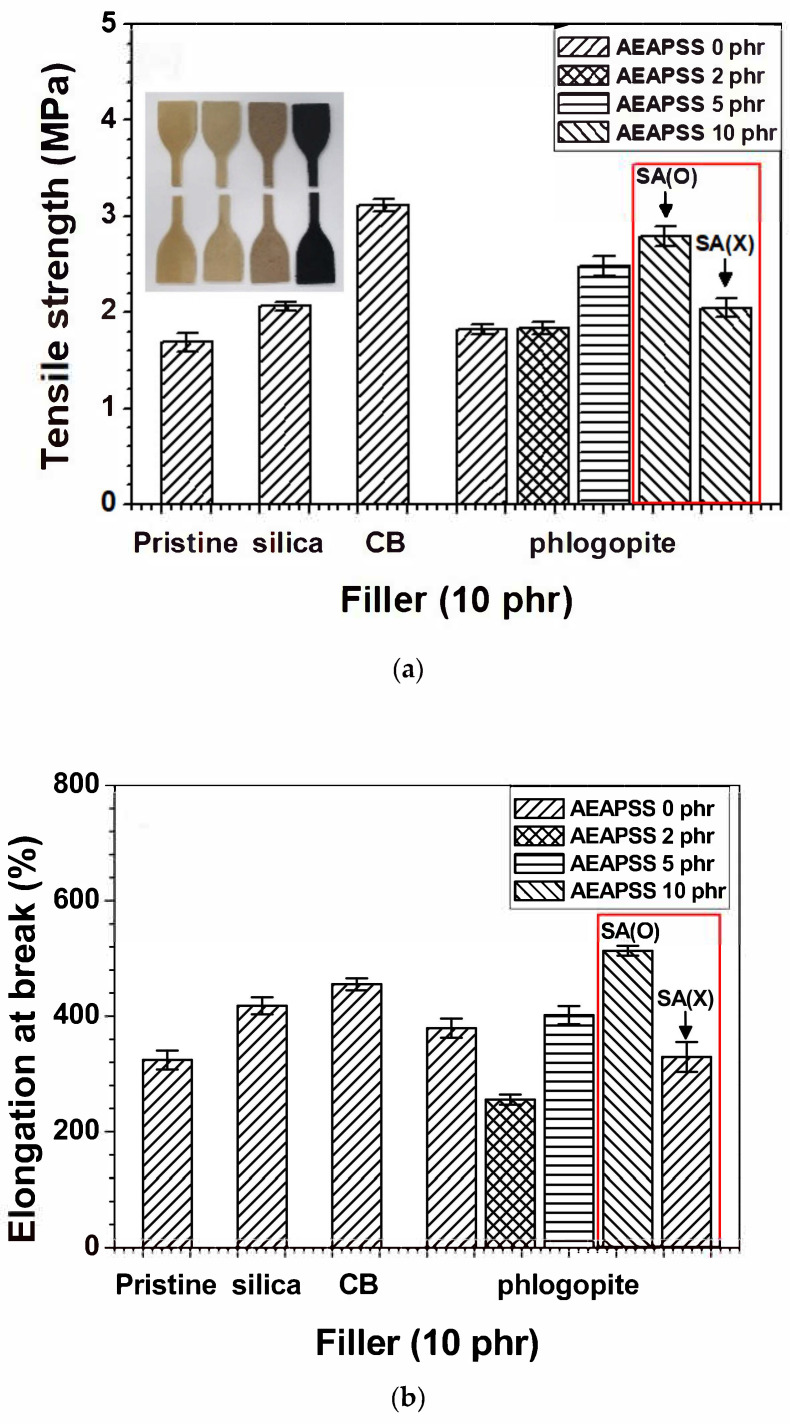
Mechanical properties of the NR/EPDM blends and composites with different fillers and AEAPSS concentrations: (**a**) tensile strength; (**b**) elongation at break; (**c**) tensile modulus at 100%; and (**d**) toughness. The inset of Figure 3a indicates the pristine NR/EPDM, silica-, phlogopite-, and CB-embedded NR/EPDM composites, from left to right.

**Figure 4 polymers-13-02318-f004:**
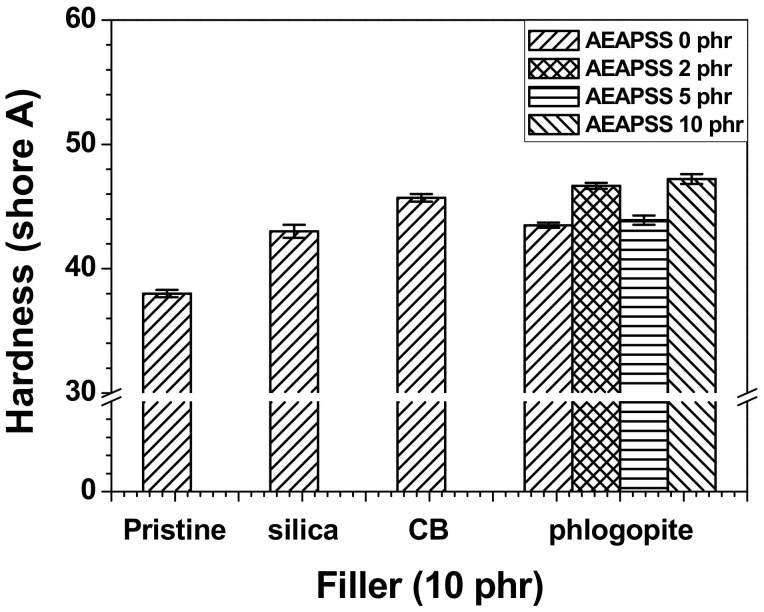
Hardness of the NR/EPDM blends and composites with different fillers and AEAPSS concentrations.

**Figure 5 polymers-13-02318-f005:**
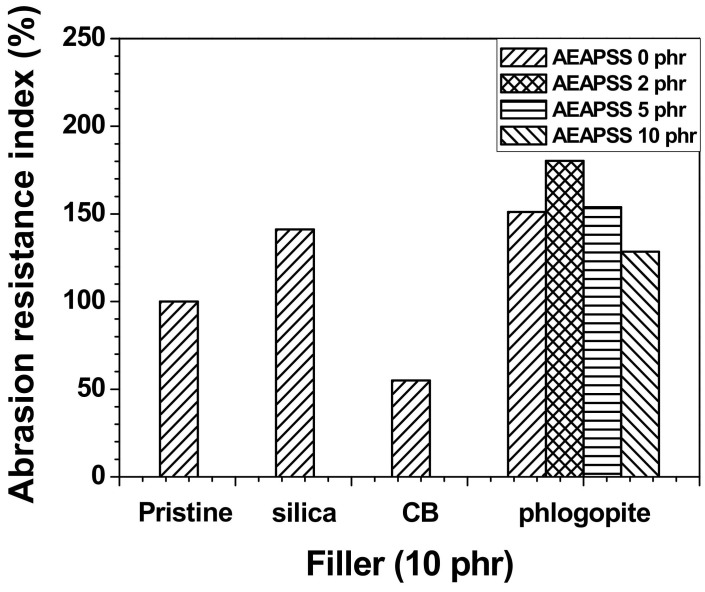
ARI of the NR/EPDM blends and composites with different fillers and AEAPSS concentrations.

**Figure 6 polymers-13-02318-f006:**
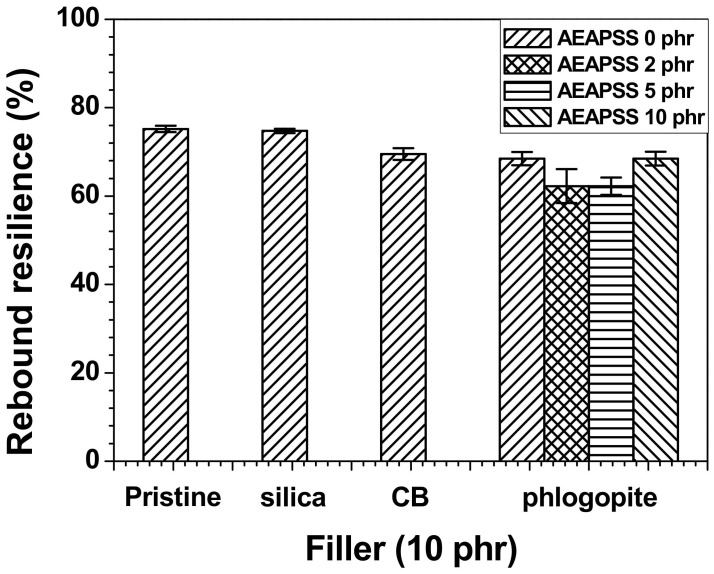
Rebound resilience of the NR/EPDM blends and composites with different fillers and AEAPSS concentrations.

**Figure 7 polymers-13-02318-f007:**
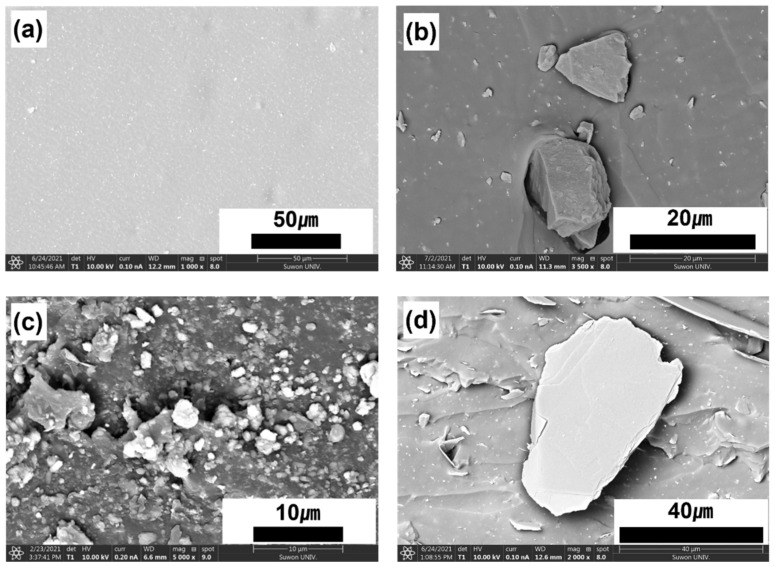
SEM images of the NR/EPDM blends and composites with different fillers and AEAPSS concentrations: (**a**) none, (**b**) silica 10 phr, (**c**) CB 10 phr, (**d**) phlogopite 10 phr, (**e**) phlogopite 10 phr/AEAPSS 2 phr, and (**f**) phlogopite 10 phr/AEAPSS 10 phr.

**Figure 8 polymers-13-02318-f008:**
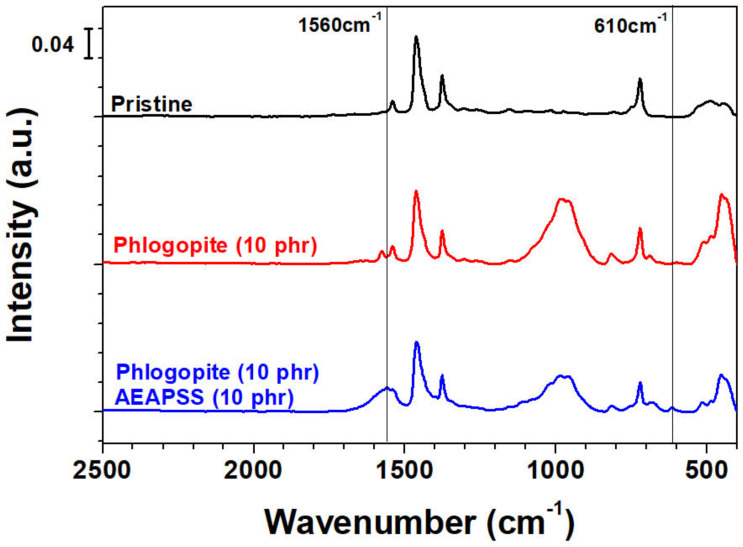
ATR-FTIR spectra of pristine rubber blend and composites consisting of phlogopite 10 phr and phlogopite 10 phr/AEAPS 10 phr.

**Figure 9 polymers-13-02318-f009:**
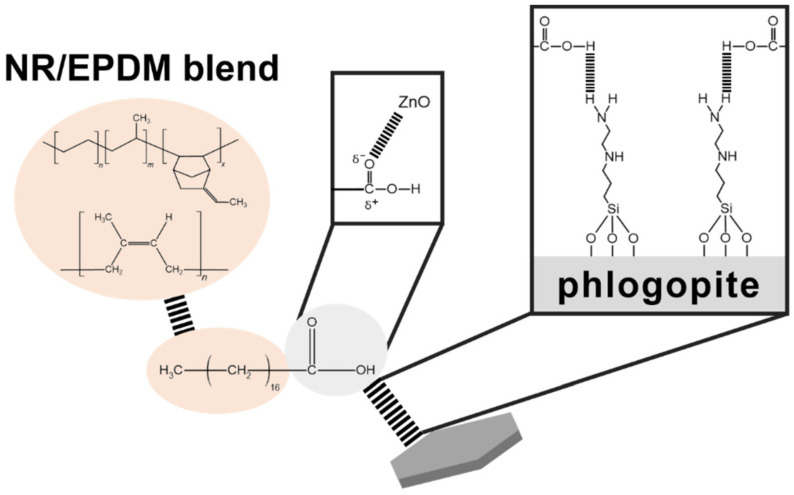
Coupling interactions among phlogopite, rubbers, and stearic acid.

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
