# Peer review of "Phlogopite-Reinforced Natural Rubber (NR)/Ethylene-Propylene-Diene Monomer Rubber (EPDM) Composites with Aminosilane Compatibilizer"

_polymers, 2021, doi:10.3390/polym13142318_

Round 1
Reviewer 1 Report
In the present work, the production and the properties of natural rubber (NR)/ethylene-propylene-diene monomer (EPDM)/phlogopite (a mica type mineral) composites are studied. More specifically, the effect of untreated and silane-treated phlogopite addition on the properties of NR/EPDM 5/3 mixture is studied and compared to the effect of silica or carbon black addition. Αs a whole it is a decent research work, but there are some points that can be improved and further highlight this effort.
First of all, the introduction could be enhanced with a paragraph on mica composite polymers (or rubbers specifically). Phlogopite may not be as common in the synthesis of polymer composites, however other types of mica have been used extensively and some reference to the effect of mica addition on the properties of composites would be informing to the readers.
As it concerns silica and carbon black fillers, some information or characterisation is necessary. What is their primary particle size? (Data from providers or SEM). SEM images of the pristine fillers could also be presented along with the rest images in Figure 7.
Moreover, some photographs of the produced composites could also be added, where the color of the samples as well as the dispersion of the fillers would be visible.
In Figure 3b, how is the lower elongation at break explained for the "AEAPS 2 phr" sample? How many specimens were tested for each sample? A valid explanation for this behavior should be provided, while the claim in line 175, that all composites have a higher elongation at break than the pristine polymer, must be changed.
Apart from SEM images of pristine silica and carbon black, a SEM image of sample "AEAPS 2 phr" in Figure 7 would be very usefull, so the dispersion of phlogopite in the polymer matrix to be examined. In addition, a comment about the grains (what are these?) in image 7a would be appreciated.
In conclusion, the behavior of most polymeric (nano)composites depends on the successful dispersion of the additives homogeneously in the polymer matrix. In this work, the dispersion of the fillers could be examined in microscopic scale via TEM or XRD (for the delamination of phlogopite). Moreover, the successful silane-modofication of phlogopite could proved via FT-IR spectroscopy. With either of these techniques the claims of the article would become more solid.
Reviewer 2 Report
This article includes meaningful findings of experimental results. However, there is a rack of explanation to be rethought as follows. I highly recommend the revision.
- The explanation of the bonding between SA and rubber as written in lines 226 and 227 and the part of its bonding in Fig. 8 are very unclear (1), whereas the bonding between SA and AEAPS is OK as shown in Fig. 8. In addition, the role of bonging by filler such as ZnO is not delineated in Fig. 8 and the explan ation dose not shown in the text ( As referring the following known findings on (1) and (2), lines 220 230 and Fig. 8 should be corrected by indicating the reference.
Concerning (1), there exists the bonding between NR and SA by the role of emulsion polymerization of SA. When the single isoprene molecule of NR is shown by A (ie. NR can be shown An ), NR is anionized as shown by the following equation, because NR involves water.
??↔(??)∗↔(?−?+)∗↔(?−?+)↔?−+ ?+
Where, because EPDM is non diene rubber, EPDM dose not contribute the bonding by ionization. Therefore, SA has role of emulsion polymerization
Concerning (2), ZnO which was used in the present article has the role of donor to become cationic easily. Therefore, the bonding between ZnO and NR occurs by ionic bond. On the other hand, t he carboxyl group of SA is ionized by mesomeric effect (ie. C=O changes C(δ+)-O( δ-). Therefore, the bonding between ZnO and O of carboxyl group occurs by ionic bond.
2. The property of tension of the rubber involving fillers changes by the tensile strength, for example, in elastic and plastic areas. Therefore, it is unclear which tensile strength the results of tensile strength in Fig. 3 indicates.

Reviewer 3 Report
The paper presents investigation of NR/EPDM bends compatibilized with various fillers with an emphasis on phlogopite. The research question is interesting and finding the answer could bring a new value to the scientific field of rubber technology. Unfortunately, the interpretation of the results is not very deep and some of the exlanations are doubtful, with no reference to the literature. The references are used only in the introduction section, no single one in the discussion! There is a number of doubtful interpretation that seems to be a risky hypotheses invented by the authors with no in-debth investigation. In this view I have to reject this paper in current form. Please find my detailed comments below:
- EPDM abbreviation shouldn’t be explained as ethylene-propylene-diene monomer, since it is already polymerized (cannot be called monomer) please explain it as ethylene-propylene-diene rubber.
- Two-roll mill – not miller (line 84)
- Is it true that the tensile sample cross-section was 15 mm x 4 mm? Or it shuld be 1.5 mm x 4 mm? (line 102)
- Please provide more details about the abrasion tester. What was the counter material? Who is the producer? What was the sample-counter sample contact geometry?
- The Mooney viscosity results are very surprising. Usually, incorporation of a filler into a rubber results in a noticeable increase of the viscosity. This is mainly because of hydrodynamic effect of a particulate filler, filler-filler and filler-rubber interactions. Even though in case of such low filler content (10 phr) the filler-filler interactions might be negligible, the hydrodynamic effect and filler-rubber interactions should increase the compounds’ viscosity.
Honestly, I doubt in the explanation that addition of 10 phr of any filler could cause such severe chain-scission of rubber molecules that would lead to a significant reduction in Mooney viscosity, especially when the compounds were prepared by two-roll mill. Did the authors find any literature sources confirming such effect? If this hypothesis is to be proposed I suggest checking the molecular mass distribution of the rubber before and after compounding. Only then such explanation should be used.
Nevertheless, the results are strange. Might it be that the samples started to age and cross-link? In fact, no anti-aging additive wea incorporated into the compounds to stabilize them. Maybe id would be advisable to redo the tests?
Also why there is no unit for the Mooney viscosity? Based on the conditions it should be ML(1+4)@125 °C, MU. - The authors claim that the significant decrease of T90 after addition of the silane is caused by stabilization of the phlogopite/rubber interface but don’t explain what the mechanism behind it is. Firstly, what does it mean that the interface is stabilized? And secondly, how this can affect the kinetics of vulcanization? It is a very unclear statement. Based on my experience introduction of amine compounds usually speeds up the vulcanization process by making the environment more basic – and I think this is what we observe in Figure 2.
- The abstract suggests that the paper presents an investigation on an effect of stearic acid on the phlogopite/rubber compatibilization but in fact the SA was added to all the samples in the same amount similarly to ZnO. Based on the results it is impossible to claim any effect of SA on the phlogopite/rubber compatibilization.
- To my knowledge toughness can be measured on materials that break with a brittle crack, in case of rubber such measurement is not feasible. How was it possible to add the toughness values to the mechanical properties of the rubber (Figure 3d)? Also, the toughness measurement details are not present in the materials and methods section.
- The scheme of interactions presented in Figure 8 is not wrong. However, to really investigate the role of SA in the compatibilizing system at least a reference sample (without SA) should be tested.
- The AEAPS was introduced in a huge overdose, up to 1:1 wt.% ratio with the phlogopite. This is a very significant amount especially for a low specific surface area filler that phlogopite is. In such systems a potential self-condensation and oligomerization of AEAPS should be taken into consideration and discussed. Especially, when the silane molecule contains two amine moiety that catalyze silanization reaction significantly.
- The SEM pictures analysis is very limited and not informative – basically the sample without AEAPS show no difference in the phlogopite delamination that the sample containing 10 phr of AEAPS.
- The authors claim supportive interaction between AEAPS, SA and the rubber but no stochiometric calculation are done – what should be the amount of SA to provide the same amount of the reactive sites as the silane for each sample.
- It would be more informative to display the full curves of the measurements instead of a single value in the column figures, like tensile or vulcanization kinetics.
The AEAPS silane containing two amino groups is not compatible with non-polar rubber like NR and EPDM, what is the reason of choosing it?
Round 2
Reviewer 2 Report
I suggest the acceptance of the publication.
Reviewer 3 Report
All my concerns were addressed. I recommend to accept the paper.